# Link Quality Estimation for Wireless ANDON Towers Based on Deep Learning Models

**DOI:** 10.3390/s22176383

**Published:** 2022-08-24

**Authors:** Teth Azrael Cortes-Aguilar, Jose Antonio Cantoral-Ceballos, Adriana Tovar-Arriaga

**Affiliations:** 1Centro de Tecnologia Avanzada, CIATEQ A.C., Jalisco 45131, Mexico; 2Tecnologico de Monterrey, School of Engineering and Sciences, Monterrey 64849, Mexico; 3Instituto Tecnologico Jose Mario Molina Pasquel y Henriquez, Jalisco 45019, Mexico

**Keywords:** wireless sensor network, link quality estimation, deep learning, failure detection

## Abstract

Data reliability is of paramount importance for decision-making processes in the industry, and for this, having quality links for wireless sensor networks plays a vital role. Process and machine monitoring can be carried out through ANDON towers with wireless transmission and machine learning algorithms that predict link quality (LQE) to save time, hence reducing expenses by early failure detection and problem prevention. Indeed, alarm signals used in conjunction with LQE classification models represent a novel paradigm for ANDON towers, allowing low-cost remote sensing within industrial environments. In this research, we propose a deep learning model, suitable for implementation in small workshops with limited computational resources. As part of our work, we collected a novel dataset from a realistic experimental scenario with actual industrial machinery, similar to that commonly found in industrial applications. Then, we carried out extensive data analyses using a variety of machine learning models, each with a methodical search process to adjust hyper-parameters, achieving results from common features such as payload, distance, power, and bit error rate not previously reported in the state of the art. We achieved an accuracy of 99.3% on the test dataset with very little use of computational resources.

## 1. Introduction

ANDON tower lamps have evolved from simple alarm systems with visual signals in the traditional factory towards a wireless sensor network (WSN) at the heart of Industry 4.0 capable of remotely identifying machine status whilst getting information about the workshop productivity [1,2]. The alarm signals for ANDON tower lamps are established by the international standard IEC 60073:2002 [3] in which the red code sends a fault or emergency stop message, the amber code sends a warning signal for operation in an abnormal process condition or machine setup, and the green code indicates a normal operation signal, whilst the blue and white codes represent user-defined signals. These types of systems are readily available on the market, such as NH-3FV2W from the company Patlite [4], TL70 from Banner Engineering Corp. [5], and SmartMonitor by Werma Signaltechnik [6].

In the industry, the information generated by WSN such as wireless ANDON tower systems is of paramount relevance because it allows for on-time error correction, applying containment measures and preventing different problems not possible to anticipate in a traditional industry. The reliability of such information is even more relevant in Industry 4.0 since it is used for different automation processes aided by the existence of intelligent, autonomous, and collaborative machines that communicate with each other in real-time to perform actions based on the environment data [7]. In this regard, machine learning models such as deep neural networks (DNNs) have made remarkable progress that may have a big impact in the industry with highly reliable analytical solutions [8].

### 1.1. Application of Machine Learning in the Industry

Deep learning (DL) is a subset of ML methods that aim to model data with complex architectures that combine different non-linear transformations. The fundamental components of DL models are artificial neural networks (ANNs), which combine to form DNNs. These techniques have enabled significant progress in the fields of image processing [9,10], pattern recognition [11] including facial recognition [12], speech recognition [13], computer vision [14], automated language processing [15], text classification [16], and recently, in the link quality estimation for a WSN. The fundamental characteristic of a DNN is the integration of multiple sequential layers, each made of multiple artificial neurons that require proper initialization and optimization algorithms for optimal performance. The right choice of DNN architecture can lead to results with outstanding performance compared to other methods [17]. In addition, the application of a multilayer ANN was shown to achieve better performance in pattern recognition in complex and highly variable environments [18]. Within the industrial sector, the multilayer perceptron (MLP) architecture is the one with the highest number of application cases partly due to its simplicity, which allows its implementation in devices with limited computational resources, such as those commonly found in the industry in developing countries [19]. The MLP has a basic architecture, where each unit or neuron in one layer is linked to all units in the next layer (fully connected) but has no link with neurons in the same layer. The number of hidden layers, the number of neurons in each layer, and the activation function define the architecture of the MLP model.

The DL paradigm has been used successfully in classification problems; however, neural networks with many layers are sensitive to weight initialization and premature learning stoppage as the gradient values decrease towards small values, the phenomenon which is known as the vanishing gradient problem [20]. To ameliorate this problem, we may use the rectified linear unit (ReLU) activation function which has a maximum gradient value of 1, without saturating, allowing the network to continue learning. For this reason, in the present paper, the ReLU function was used as the activation function of the MLP model.

With the continuous increase in device connectivity and computing capabilities of digital systems, some industries have succeeded in incorporating new methods based on artificial intelligence (AI). Hence, data provided by a wide variety of sensors are used to detect and prevent a possible failure or breakdown, especially via machine and process supervision systems. Indeed, failure detection is critical in the industry because it allows determining if it is necessary to stop a process and carry out maintenance actions [21]. Márquez-Vera et al. [22] use a DL model with long short-term memory (LSTM) in a failure detection case. They reported that the neural network was more sensitive to hyper-parameters such as batch size or the learning rate regarding the number of layers, which increased until no improvement was found.

### 1.2. Motivation

In today’s industry, the ANDON tower works as a sensor node in the WSN that sends an alarm signal without information about data reliability. After a comprehensive literature survey, we found that this alarm signal in conjunction with link quality classification has not been used in applications for wireless ANDON towers in the industry. In this paper, we propose a novel application for a deep neural network model based on a multilayer perceptron (MLP) to predict the link quality estimation (LQE) and consequently, the reliability of the information used for decision making in the workshop. In addition, the LQE can also be used to improve WSN performance.

The packet reception rate (PRR) was used to define the LQE classes, and 10,000 data points were gathered from a realistic experimental scenario using actual industrial machinery. Using these data, we carried out comprehensive experimentation comparing different machine learning (ML) models, including random forest (RF), support vector machines (SVM), K-nearest neighbors (KNN), gradient boosting (GB), and MLP, which displayed the best performance from all the models, achieving an accuracy >99% on the test dataset after a thorough hyper-parameter tuning. In order to choose the most suitable learning rate, we carried out extensive random searches in the range of 3 × 10^−8^ ≤ LR ≤ 3 × 10^−2^ until finding that an LR = 2.95 × 10^−5^ achieved the best performance in the training and validation sets. In spite of the model’s simplicity, MLP achieved high precision in classification tasks, making it the ideal model for our industrial application due to its low computational requirements both for training and inference.

The remainder of this paper is organized as follows. Section 2 presents a review of previous work related to our research. Then, Section 3 refers to materials and methods and describes the hardware, MLP model, and dataset, and Section 4 shows the results and comparison to other machine learning models. Finally, Section 5 and Section 6 present the discussion and conclusions, respectively.

## 2. Related Work

Machine monitoring plays an important role in the industry due to its effects on production quality and production cost. It can be divided into two types, machine condition monitoring and machine process monitoring. The first one refers to the monitoring of machine components, such as gears and bearings, and the second one refers to monitoring workpieces and tools. The first step in machine monitoring is signal acquisition, for example: vibration, temperature, cutting force, product labelling, and others. In this context, WSNs are the key to achieving advanced automated systems by turning traditional machines into modern machines. The synergy between WSN and machine learning allows intelligent decision making and LQE estimation with efficient computational time and high accuracy [23].

The propagation environment of radio frequency signals can change in space and time and affect the quality of the link in WSN. Consequently, to achieve reliable and error-free transmission, the transceiver’s parameters must be adjusted depending on the best LQE estimate. Over the last few years, machine learning algorithms have been used to estimate LQE due to their ability to process and learn from large amounts of data, which can be gathered from various scenarios and technologies [24]. 

The link quality metrics for LQE algorithms fall mainly into three categories: software, hardware, and scores [25]. The software LQE metrics such as packet reception rate (PRR), required number of packet retransmissions (RNP), and expected transmission count (EXT) are calculated using the transmission variables. The PRR measurement is the ratio of the number of packets successfully received to the number of packets transmitted within a time window, but PRR-based algorithms are not sensitive to rapid fluctuations in link quality [26]. The RNP measurement is the average number of packet transmissions and retransmissions required before a successful reception; nonetheless, it is unstable and it cannot estimate packet delivery for asymmetric links successfully [27]. The EXT measurement refers to the number of transmissions required to send a complete packet; however, its implementation often fails in congested networks [28]. 

The hardware LQE metrics such as received signal strength indicator (RSSI), link quality indicator (LQI), and signal to noise ratio (SNR) are acquired from the transceivers. The RSSI is the measurement of the intensity of the signal present in the receiver that evaluates the quality of the link via supervised learning algorithms, and it classifies the quality of a link as high or low for a given routing protocol. However, the RSSI indicates only the power of the received signal at the receiver, and it does not provide an accurate measure that considers the affectation due to noise. The LQI measurement is a characterization of the strength or quality of a received packet according to the IEEE 802.15.4 standard; however, the algorithms based on LQI have three disadvantages: The variance of the LQI increases significantly as the quality of the link degrades;There is no consensus among transceiver manufacturers on how to calculate LQI;It is only useful for transceivers that work with the IEEE 802.15.4 standard [29].

The SNR metric is the difference, in decibels, between the received signal strength and the floor noise. Despite algorithms based on SNR measurement being easy to implement, these are not suitable for random SNR behaviors, and these are unable to provide an accurate description of link quality [30].

The score algorithms to calculate the LQE metric use various input parameters; for example, Luo et al. [31] used SNR, LQI, and RSSI of up and down links in a neural network architecture of seven stacked layers to extract information from the asymmetric characteristics of the input data. Then, the classification was made by an SVM algorithm. This algorithm is focused for bidirectional communications and it is not suitable for unidirectional communications, such as the application presented here.

The use of physics and software metrics such as SNR, RSSI or PRR allows calculating the instantaneous LQE parameter without considering its historical trend. In a recent research, He and Shu [32] proposed an LQE deep forest algorithm that estimates the PRR from the input of the mean of LQI, RSSI, and SNR metrics and stability parameters CV_LQI_, CV_RSSI_, and CV_SNR_ that the authors calculated as the standard deviation divided by the mean. Abdel-Nasser et al. [33] predicted the LQE using long short-term memory (LSTM) and gated recurrent units (GRUs). However, the implementation of these algorithms has the disadvantage of needing large amounts of historical data for networks that are already in operation.

### Contribution

The most widely used methods to predict the link quality of WSN use input metrics acquired from the transceiver such as RSSI, LQI, and SNR or metrics calculated by software such as PRR, RNP, and EXT. Other algorithms seek to improve prediction accuracy using many metrics, historical data, and statistical trends. However, despite the reported successful results, some of these metrics cannot be easily acquired from the transceiver or are not readily available. In addition, the quality of a link can also be affected by other factors such as the distance from the transmitter to the receiver, the transmission rate, the data payload, the transmitter’s power, and the presence of noise and interference, as well as the random presence of obstacles and objects that can attenuate, reflect, or scatter the radio frequency signal, consequently increasing the bit error rate (BER). 

For complex industrial environments, the application of a DNN classification algorithm is a better choice to predict LQE than other traditional methods based on statistical models and fuzzy logic [34,35]. Moreover, the MLP model proposed in this paper compared to other machine learning models can help adjust the best WSN parameters, such as distance, transmission rate, power, and data payload in the transceiver. In addition, our work differs significantly from the current state of the art because, at the moment, there is no industrial solution to predict the LQE and the associated information reliability with alarm signals produced by modern wireless ANDON towers. The application of this technology will further the digitalization and evolution from a traditional factory to a smart factory within the Industry 4.0 model, particularly in developing countries, where the incorporation of new technology is slow and difficult.

## 3. Materials and Methods

In a WSN with a star network topology, communications occur between two types of devices, a coordinator node that manages those with the lower nodes, and the lower nodes that transmit the signal from the sensors to the coordinator node, as shown in Figure 1.

To describe our industrial WSN setup, this section is divided into seven parts; the first one describes the transceiver used in the ANDON tower that works as a sensor node; the second one describes the characteristics of the computer equipment used for the programming of the MLP model; the third one describes the experimental scenario where data were collected; the fourth one addresses the neuronal network’s features, i.e., the model’s input variables; the fifth one is the LQE classification based on PRR, i.e., the model’s output variables; the sixth describes the dataset structure; and the seventh one explains the MLP model structure.

### 3.1. Transceiver

The ANDON tower uses the low-cost NRF24L01 [36] transceiver to send the experimental data from the sensor node to the coordinator node, with a computer connected to the coordinator node to store the data in a CSV file. The transceiver works in the 2.4 GHz free band and allows setting the transmission power at −18 dBm, −12 dBm, −6 dBm, and 0 dBm. The transmission rate setting options are 250 kbps, 1 Mbps, and 2 Mbps. The transceiver uses the SPI protocol for communication with the ANDON tower microcontroller. Its maximum range is 30 m, which is sufficient for the maximum distance in our experimental scenario of 16 m. The communication tests were realized by sending a payload of 2, 4, 8, 16, and 32 bytes.

### 3.2. Computer Hardware and Software

The MLP classification model was developed using the PyTorch library [37] and ran on a GPU, whilst the other ML models were developed using the SkLearn library [38]. All the models were trained on a standard laptop equipped with INTEL core i7 CPU, 8GB of RAM, and an NVIDIA GEFORCE 940MX graphics card. PyTorch is a library developed by Meta (formerly Facebook) that facilitates the construction of deep learning projects through models coded in Python. For the scientific community, PyTorch has become an outstanding deep learning tool due to its accessibility, ease of use, and wide range of applications [37].

### 3.3. Experimental Scenario

The main challenge to implement a wireless ANDON tower for industrial monitoring and control systems is satisfying reliability and performance requirements, where some accepted indicators for the reliability of the physical layer in each node are the quality of the wireless link, based on PRR and BER [39]. Figure 1 shows the experimental scenario where the communication tests of the ANDON towers were carried out and where experimental data were acquired. The ANDON tower worked as the sensor node in the WSN, with its transceiver programmed to send 100 test data packets to the receiver node that worked as the coordinator node. The receiver node calculated the PRR from the correctly received packets. Data were saved in a computer connected to the coordinator node. The receiver-coordinator node was placed in a fixed position in the workshop, while the location of the sensor-transmitter node was changed according to the distances selected for the experiment. A star topology was used in the network.

### 3.4. Neuronal Network’s Features

Our classification model to predict the LQE was trained on an original dataset with the following features: the transmission rate x_1_, the transmitter power x_2_, the distance from the transmitter to the receiver x_3_, the payload x_4_, and the bit error rate x_5_. Table 1 shows statistical information of the input data. The data rate, power, and payload were collected from the transceiver configuration. Distance data were collected by locating the sensor node at different positions in the experimental scenario and the bit error rate (BER) was calculated with (1), where *f = 8n* is the length of the packet in bits, and n is the data payload [40].
(1)$$\mathrm{BER}=1-{\mathrm{PRR}}^{\left(f^{-1}\right)}$$

### 3.5. LQE Classification Based on PRR

The PRR measurement is used as a classification parameter in the LQE prediction model [31,32,33,34,35]. In the experimental scenario presented in Figure 1, the transmitting node periodically sent 100 test packets to the receiver node that calculated PRR. The LQE was related to the rate of packets received successfully. Forecast categories were divided according to (2). We considered that a PRR smaller than 0.89 is not acceptable for our industrial application; 2500 random samples were collected for each of the four LQE categories.
(2)$$y=f_{\left(\mathrm{PRR}\right)}=f\left(x\right)=\{\begin{array}{ccc} 1, & Best & \;\mathrm{PRR}=1\\ 2, & Good & \;0.96<\mathrm{PRR}\leq 0.99\\ 3, & Common & \;0.89<\mathrm{PRR}\leq 0.96\\ 4, & Bad & \;\mathrm{PRR}\leq 0.89 \end{array}$$

Figure 2, Figure 3, Figure 4 and Figure 5 show the scatterplots of the experimental measurements for distance, transmission rate, power, and data payload against BER. The LQE classification is denoted by numbers in the scatterplots, where the best subset is denoted by number 1, good by number 2, common by number 3, and bad by number 4, as explained in (2). The BER observation range in the scatterplots was chosen to identify the four forecast categories, and we can see that the BER mean value was low because experimental data were gathered with low noise working conditions, i.e., from (1), we can see that BER = 0 when PRR = 1. The scatterplot in Figure 2 shows that BER decreased with distance. In Figure 3, we did not see a strong relationship between transmission rate and the BER variable; however, forecast categories 1 and 2 had low BER values. On the other hand, increasing the transceiver’s power improved the quality of transmission with better BER values, i.e., more data packets without errors arrive at the receiver, whilst data from categories 1 and 2 tended to have lower BER values, as shown in Figure 4. For the NRF24L01 transceiver, a high payload increased the transmission quality, and finally, Figure 5 shows a clear division between forecast categories. We selected values of rate, payload, and power based on the NRF24L01 transceiver characteristics.

### 3.6. Dataset

From the experimental raw data, we selected 10,000 random data. In order to avoid the problem of class imbalance [35], 2500 random samples were selected for each of the four LQE categories. Then, 80% of the dataset was used for training, 10% for validation, and 10% for testing. A mini-batch size of 2048 data points was chosen with a random shuffling method.

Figure 6 shows the dataset’s structure. The input variables data rate, power, and payload depend on the transceiver’s setting options, in contrast to the distance and BER that depend on experimental measurements.

### 3.7. Estimation Model

The multilayer perceptron is one of the most established deep learning models for non-linear classification and regression. These architectures are frequently used in the industry to model systems and predict phenomena due to their ease of implementation with relatively low computational requirements for training compared to other more complex methods [41]. In addition, an MLP is a suitable option for small workshops in developing countries such as Mexico, where computational resources are limited and commonly older than 10 years. The MLP estimation model proposed in the present paper has three hidden layers, as shown in Figure 7, with the ReLU as the activation function to avoid the vanishing gradient problem. The inputs to our MLP model were the transmission rate, the transmitter power, the distance from the transmitter to the receiver, the payload, and the bit error rate.

In order to select the most suitable learning rate, we realized two random searches. The first search was performed in a broad range of 3 × 10^−8^ ≤ LR ≤ 3 × 10^−2^ for 5000 epochs, as shown in Figure 8, where we found the best LR = 1.65 × 10^−5^; then, we repeated the process within a finer range of 1.55 × 10^−5^ ≤ LR ≤ 3 × 10^−5^; this time, the model was trained for 10,000 epochs (Figure 9). In the first search, we used the confusion matrix to evaluate the model (Figure 8), and in the second search, we used the metric of accuracy against epochs (Figure 10). The accuracy metric was calculated by (3) where TP indicates true positives and TN indicates true negatives.
(3)$$Accuracy=\frac{TP+TN}{Total}$$

## 4. Results

In this section, we present the results of the tuning process of the MLP model (see Table 2) and the comparison against other machine learning models such as SVM, RF, KNN, and GB. Experiments on the proposed dataset demonstrated satisfactory results. In order to find the best hyper-parameters, the performance of the model was evaluated by searching LR values, i.e., (a) LR = 3 × 10^−8^, (b) LR = 1.65 × 10^−7^, (c) LR = 3 × 10^−7^, (d) LR = 1.65 × 10^−6^, (e) LR = 3 × 10^−6^, and (f) LR = 1.65 × 10^−5^, as shown in Figure 9. We can see that the MLP model showed strong performance for a value of LR = 1.65 × 10^−5^, achieving an accuracy of 0.996 for the training dataset.

Although the previous results were reasonable, we carried out a second search for a finer learning rate tuning in the range of 1.55 × 10^−5^ ≤ LR ≤ 3 × 10^−5^. The MLP model achieved an accuracy of 0.993 with LR = 1.55 × 10^−5^; nonetheless, increasing the learning rate to LR = 2.95 × 10^−5^ led to an accuracy equal to 1 for the training data and 0.998 for the validation set (Figure 11). Figure 12 shows that the MLP model with LR = 2.95 × 10^−5^ presented neither underfitting nor overfitting issues.

### Comparison with Other Machine Learning Models

In this section, we present comparative experimental results to demonstrate the validity and effectiveness of the proposed MLP model over other ML alternatives. The random forest (RF) classifier was tuned with a depth of 3 and 100 estimators, keeping default values for other parameters. From the experimentation, we noticed that although increasing the maximum depth of the tree improved the model’s accuracy, it was still below than the accuracy of other models. Then, we tried with a support vector machine (SVM) classifier with a hinge loss function and trained for 2000 iterations; here, we noted that an increase in the number of iterations did not lead to better accuracy. We also tested a K-nearest neighbors (KNN) model, where we found that k = 3 led to the best results, since larger values of k led to lower accuracies. The next model we tried was a gradient boosting (GB) classifier, which built an additive model that led to very good results. To achieve this, it was tuned with 100 estimators and a maximum depth of 3. The best accuracy was achieved with a learning rate of 0.0125, with higher learning rates leading to underfitting. Figure 13 and Figure 14 show the comparison between the proposed model and the other machine learning alternatives all coded using the SkLearn library [37]. From this experimentation, we concluded the following:The MLP model had the best performance for the training, validation, and test sets, as shown in Figure 13 and Figure 14.Although the MLP achieved the best results, the GB model achieved an accuracy only 0.4% lower than that achieved by the MLP.The MLP model required more time to complete the training stage than other simpler models; nonetheless, once trained, the time it took to make the estimation with the test data was very brief, approximately 0.02 s, as shown in Figure 15.We demonstrated that the MLP model can be implemented with limited computational resources and with data gathered from a low-cost transceiver. Despite these disadvantages, the MLP model was capable of making accurate estimations. In addition, for industries in developing countries, the application of these technologies can help the transition into Industry 4.0.

## 5. Discussion

The MLP model displayed better accuracies than other ML models; nonetheless, it required more time to complete its training process. However, it should be noted that its time performance was in the same order of that of the other models when doing inferences as tested during our experimentation. This is important for a real application in the industry because it is possible to adjust the transceiver’s parameters and location inside the workshop with the aim to increase the reliability of the information sent by the ANDON towers to the coordinator node. Additionally, reliable metrics of LQE could be used to evaluate the configuration of a sensor node that works in a real environment. We considered that a PRR less than 0.89 is not acceptable for this application, and we selected LQE classes between 0.89 and 1. If LQE was equal to 1, then the alarm signal sent by the ANDON tower reached higher reliability levels. We used a novel dataset comprised by different input data types, collected from the transceiver setting option and experimental measurements. However, some input features to MLP model were restricted to the parameters allowed by the transceiver and physical constraints of our setup. As a future work, we suggest collecting more data from larger distances and payload measurements.

The alarm signal used in conjunction with LQE is a novel application for wireless ANDON towers, with the potential to ameliorate problems faced by factories located in developing countries through the incorporation of new technology during transition into the Industry 4.0 paradigm.

## 6. Conclusions

In the industry, the information of alarm signal in conjunction with the LQE class is relevant and represents a new approach to evaluate data’s reliability. In this paper, we proposed a novel application of a deep learning MLP model for the estimation of the LQE metric for a WSN integrated by ANDON towers that sends wireless information about alarm signals or about the operational status of a workshop’s machinery.

The random dataset was gathered from a realistic experimental scenario and split into 80% for training, 10% for validation, and 10% for testing. In order to find the best learning rate parameter, we realized a methodical tuning process for the MLP model and noted that a learning rate of LR = 2.95 × 10^−5^ achieved an accuracy of 99.8% for validation data without overfitting the training set.

The MLP model achieved an accuracy of 99.3% for test data and showed better performance than other machine learning models such as SVM, RF, KNN, and GB. Despite the MLP training time, its time to carry out estimations was in the order of ms, i.e., the same order as that of the other models. From our research and experimentations, we could see that despite its simplicity, the MLP is a suitable model for workshops in developing countries with limited computational resources.

## Figures and Tables

**Figure 1 sensors-22-06383-f001:**
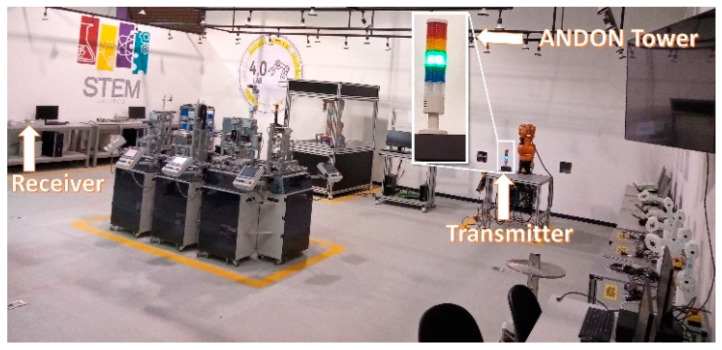
Experimental scenario, workshop TecMM, Zapopan, México.

**Figure 2 sensors-22-06383-f002:**
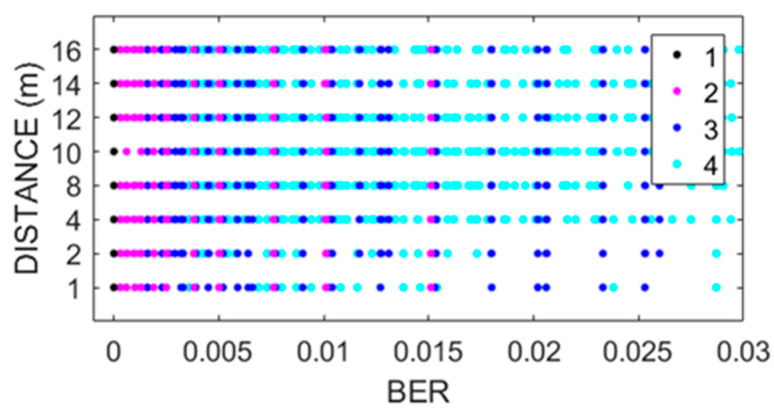
Scatterplot of distance against BER.

**Figure 3 sensors-22-06383-f003:**
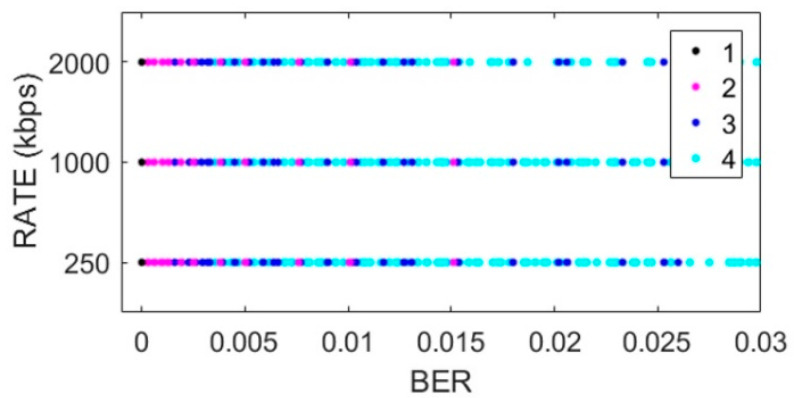
Scatterplot of transmission rate against BER.

**Figure 4 sensors-22-06383-f004:**
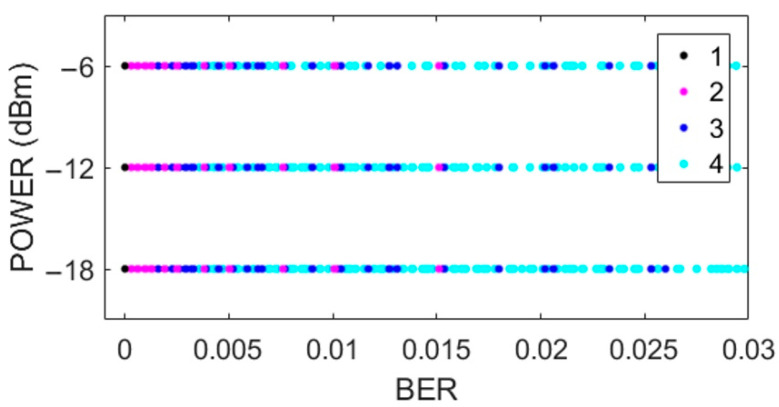
Scatterplot of transmission power against BER.

**Figure 5 sensors-22-06383-f005:**
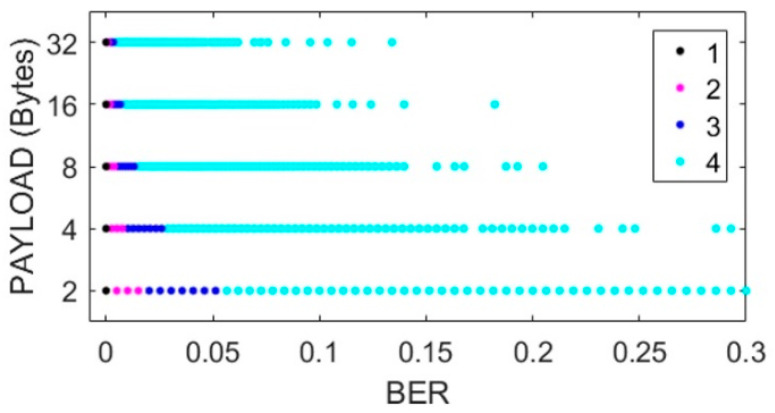
Scatterplot of payload against BER.

**Figure 6 sensors-22-06383-f006:**
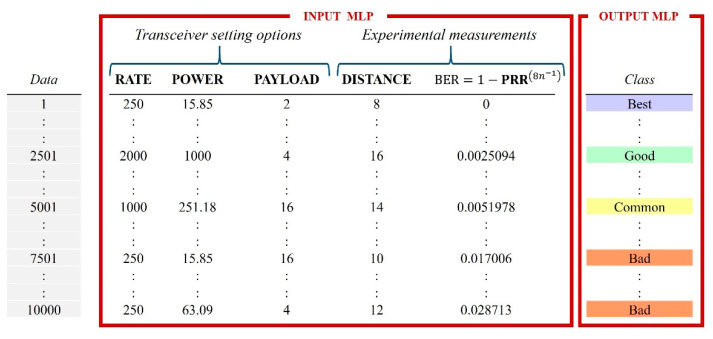
Dataset structure from CSV file.

**Figure 7 sensors-22-06383-f007:**
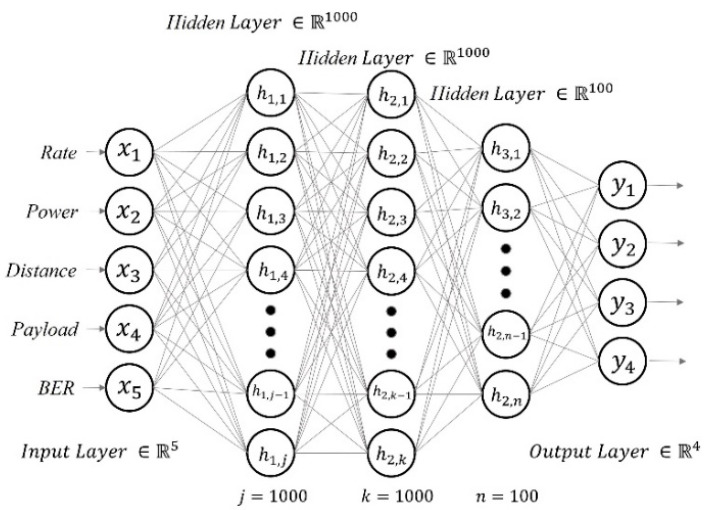
The MLP model structure.

**Figure 8 sensors-22-06383-f008:**
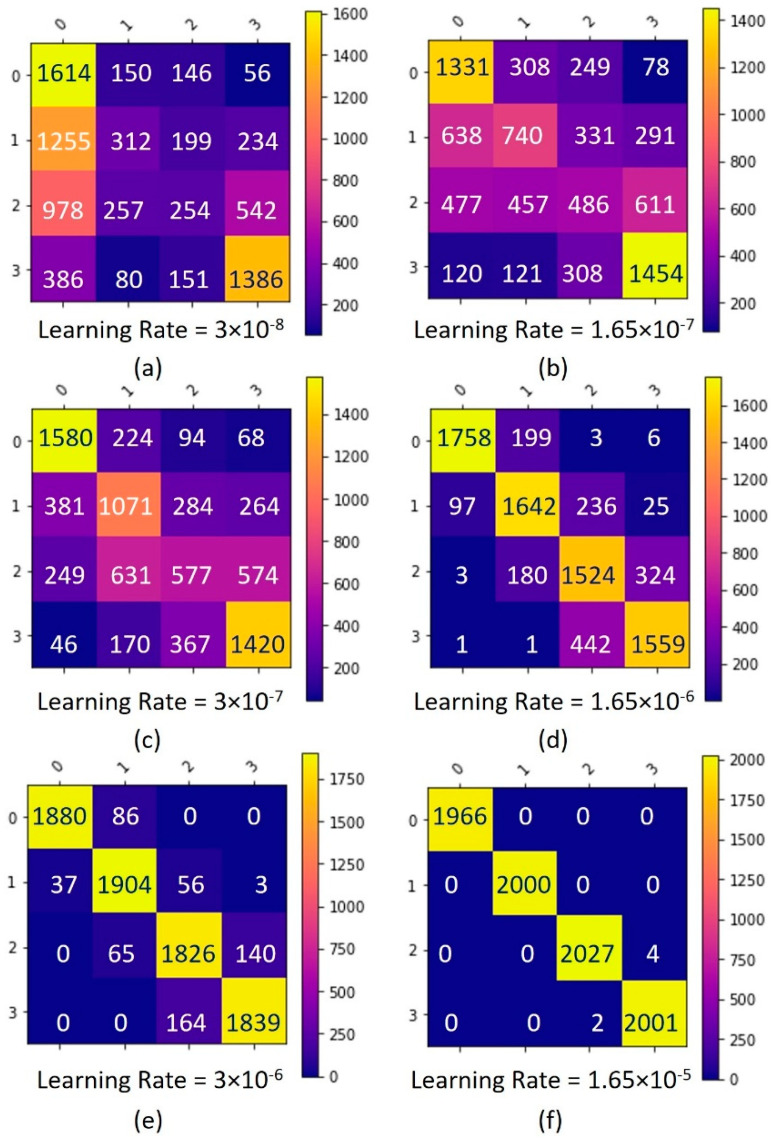
Confusion matrix for first search learning rate tuning.

**Figure 9 sensors-22-06383-f009:**
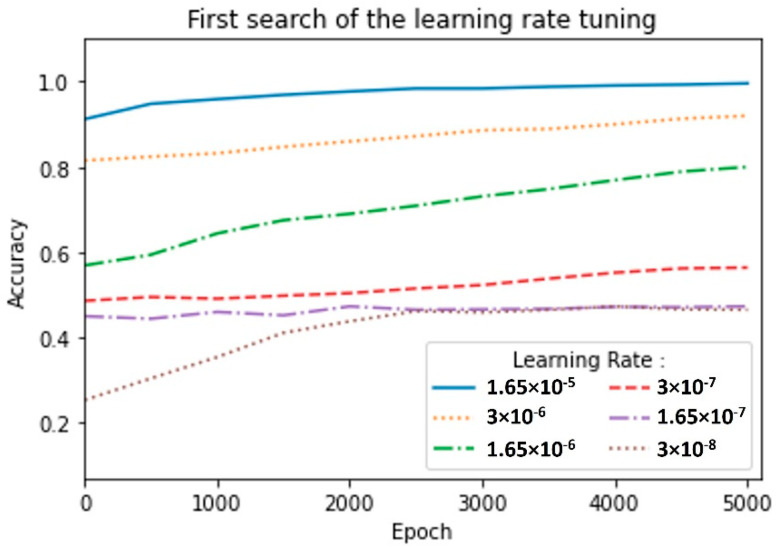
Accuracy score for training dataset with MLP model during first learning rate search.

**Figure 10 sensors-22-06383-f010:**
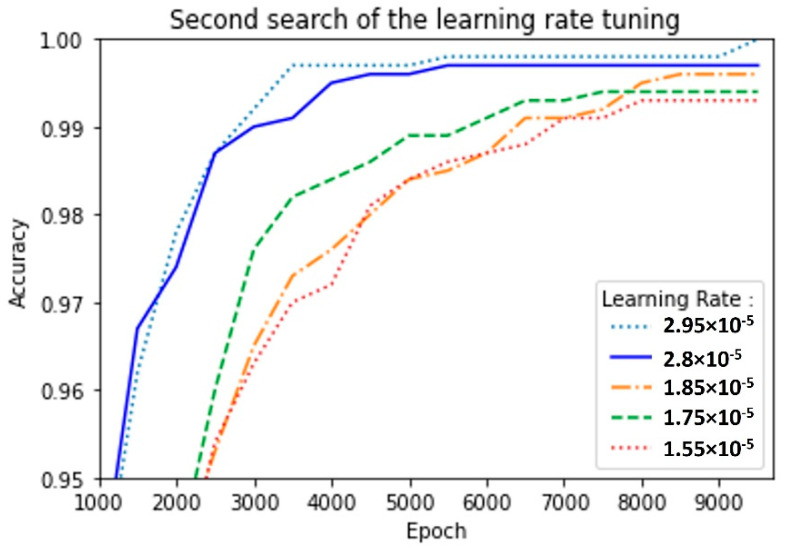
Accuracy score for training dataset with MLP model during second learning rate search.

**Figure 11 sensors-22-06383-f011:**
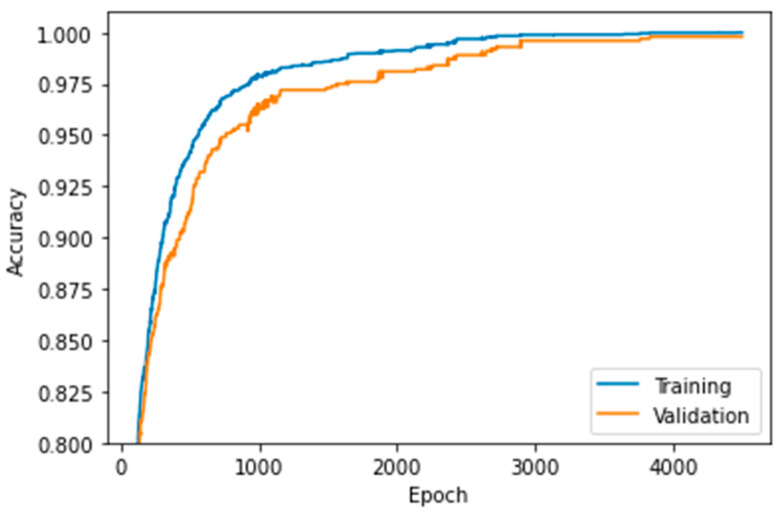
Accuracy vs. epochs for training and validation data with LR = 2.95 × 10^−5^.

**Figure 12 sensors-22-06383-f012:**
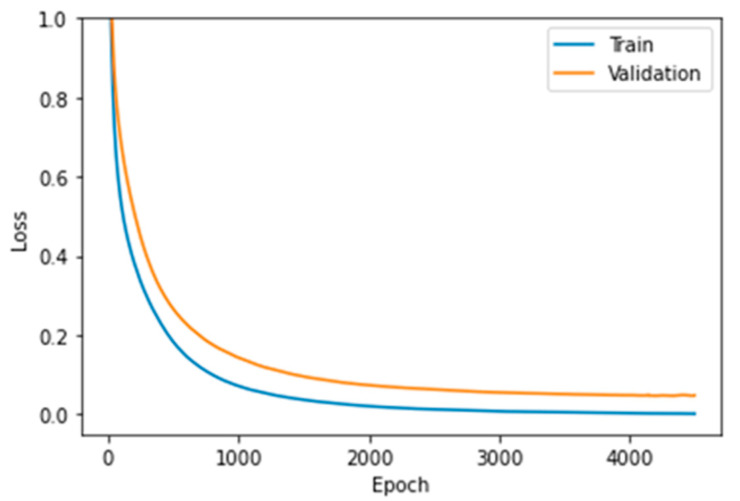
Loss function vs. epochs for training and validation data with LR = 2.95 × 10^−5^.

**Figure 13 sensors-22-06383-f013:**
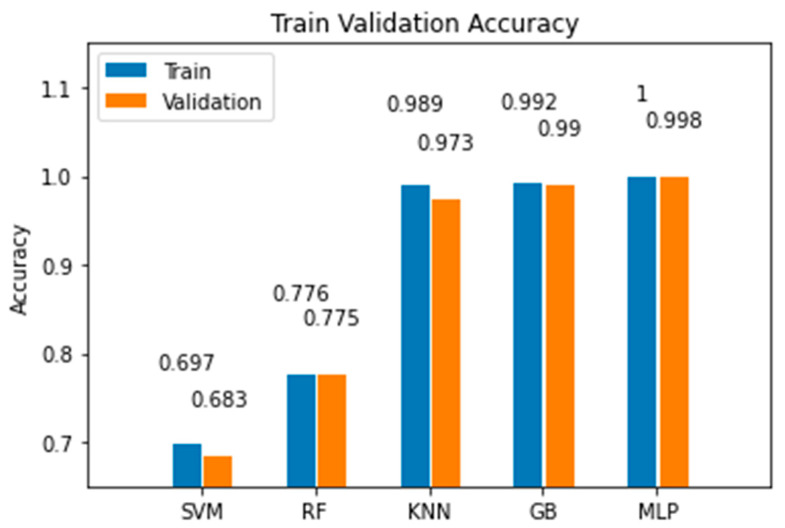
Comparison of the accuracy achieved for the training and validation data by the MLP model against other machine learning models.

**Figure 14 sensors-22-06383-f014:**
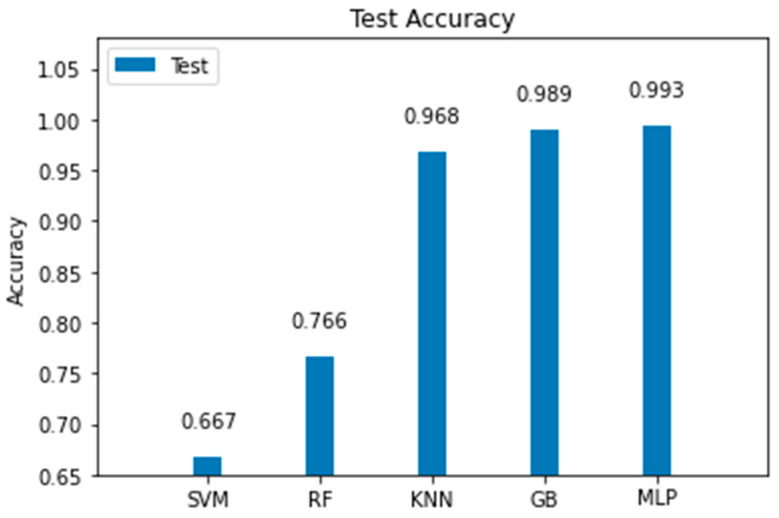
Comparison of the accuracy achieved for the test data by the MLP model against other machine learning models.

**Figure 15 sensors-22-06383-f015:**
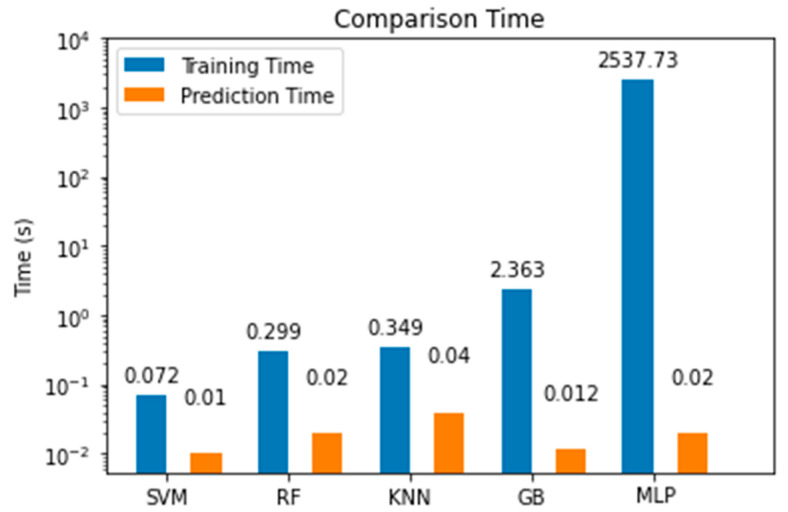
Comparison of training and estimation time for the MLP model against other machine learning models.

**Table 1 sensors-22-06383-t001:** Statistical information of the input data.

	Rate $x_{1}$	Power $x_{2}\;$	Distance $x_{3}\;$	Payload $x_{4}\;$	BER $x_{5}\;$
count	10 k	10 k	10 k	10 k	10 k
mean	1101.9 kbps	272.9 mW	8.9 m	13.9 bytes	0.0144
std	699.2 kbps	376.7 mW	4.4 m	11.4 bytes	0.0328
min	250 kbps	15.85 mW	1 m	2 bytes	0.0
max	2 Mbps	1000 mW	16 m	32 bytes	0.470

**Table 2 sensors-22-06383-t002:** Search of learning rate and accuracy score for training data.

Search	Epochs	Learning Rate	Training Accuracy
First Tuning	5000	3.00 × 10^−8^	0.465
	5000	1.65 × 10^−7^	0.473
	5000	3.00 × 10^−7^	0.564
	5000	1.65 × 10^−6^	0.800
	5000	3.00 × 10^−6^	0.920
	5000	1.65 × 10^−5^	0.996
Second Tuning	10,000	1.55 × 10^−5^	0.993
	10,000	1.75 × 10^−5^	0.994
	10,000	1.85 × 10^−5^	0.996
	10,000	2.05 × 10^−5^	0.997
	10,000	2.80 × 10^−5^	0.999
	10,000	2.95 × 10^−5^	1.000

## Data Availability

The dataset used for this research is publicly available at: https://ieee-dataport.org/documents/wireless-andon-tower-nrf24l01 (accessed on 9 March 2022).

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
