# Peer review of "Link Quality Estimation for Wireless ANDON Towers Based on Deep Learning Models"

_sensors, 2022, doi:10.3390/s22176383_

Round 1

Reviewer 1 Report

1. In this paper, the authors propose a deep learning model to reduce expenses due to early failure detection and problem prevention. They carried out extensive data analyses using a variety of machine learning models to adjust hyper-parameters and determine common features such as payload, distance, power, and bit error rate.

2. The main research contribution of this paper is the employment of deep learning to provide results regarding performance parameters such as payload, distance, power, and bit error rate. Also, a set of comparisons of many machine learning models is provided. The contribution is original with respect to the state of art. 

3. The paper well situates itself with respect to the works in the literature. The contribution of the authors is well demonstrated by the comparative study provided in the text.

4. The paper is well written. However, it requires some amendments to be made on it:

• Some language mistakes need to be fixed. E.g.: many long sentences in the abstract. A complete proofreading is needed.

5. The conclusion well summarizes the main findings of the paper and goes in conjunction with the set objectives at the introduction section.

6. The list of references is satisfactory. Yet, it may be blended by some reference items. Examples include:

• Messali, Z., SAAD SAOUD, S. and LAMRECHE, A. 2021. Covid-19 Images and Video Denoising Algorithms Based on Convolutional Neural Network CNNs. Algerian Journal of Signals and Systems . 6, 2 (Jun. 2021), 122-129. DOI:https://doi.org/10.51485/ajss.v6i2.126.

• HATHAT, A. 2021. Design of Microstrip Low-pass and Band-pass Filters using Artificial Neural Networks . Algerian Journal of Signals and Systems . 6, 3 (Sep. 2021), 157-162. DOI:https://doi.org/10.51485/ajss.v6i3.134.

Author Response

Dear Reviewer

Thank you for giving us the opportunity to submit a revised version of the manuscript “Link Quality Estimation for wireless ANDON towers based on Deep Learning Models” for publication in the Sensor Journal. We appreciate the time and effort that the reviewers dedicated to providing feedback on our manuscript and we are grateful for the insightful comments that led to valuable improvements to our paper.

We have fully revised our manuscript and incorporated the suggestions kindly made  by the reviewers. Below we provide, in blue, the point-by-point responses. All page numbers refer to the revised manuscript file with tracked changes enabled.

We hope that the revisions in the manuscript and our accompanying responses will be sufficient to make our manuscript suitable for publication in Sensor Journal.

Reviewer Points:

1. In this paper, the authors propose a deep learning model to reduce expenses due to early failure detection and problem prevention. They carried out extensive data analyses using a variety of machine learning models to adjust hyper-parameters and determine common features such as payload, distance, power, and bit error rate.

2. The main research contribution of this paper is the employment of deep learning to provide results regarding performance parameters such as payload, distance, power, and bit error rate. Also, a set of comparisons of many machine learning models is provided. The contribution is original with respect to the state of art.

3. The paper well situates itself with respect to the works in the literature. The contribution of the authors is well demonstrated by the comparative study provided in the text.

5. The conclusion well summarizes the main findings of the paper and goes in conjunction with the set objectives at the introduction section.

Reply: Thank you for your comments.

Reviewer Point:  
4. The paper is well written. However, it requires some amendments to be made on it:  Some language mistakes need to be fixed. E.g.: many long sentences in the abstract. A complete proofreading is needed.

Reply: Thank you for pointing this out. We have fully revised the entire paper, incorporating your suggestion in the abstract adding punctuation marks. Likewise, we corrected typos such as those in lines 88, 171, and 260 (just to mention some examples, the whole paper was fully proofread).

Reviewer Point:
5. The list of references is satisfactory. Yet, it may be blended by some reference items. Examples include.

Reply: As suggested by the reviewer, we have added reference 10 “A. Covid-19 Images and Video Denoising Algorithms Based on Convolutional Neural Network CNNs” and cited it in line 54, in section 1.1. Application of machine learning in industry.

Reviewer 2 Report

The paper proposes the use of ANDON towers in industrial context for link quality estimation in wireless sensor networks. The work is interesting and generally well written. However, there are some issues to be solved:

1.       English needs polishing. There are some typos and grammar mistakes. See for examples: a) line 88: it is written “work” instead of “works”; b) line 161: it is written “propose predict LQE”; c) line 249: it is written “In the experimental scenario, Figure 1” instead of ““In the experimental scenario presented in Figure 1”; d) line 314: it is written “epoch” instead of “epochs”; etc. Please check the entire manuscript.

2.       Related Work section needs to be extended with a paragraph related to the use of other types of sensor nodes to estimate the link quality within WSN;

3.       Line 183: the authors speak about “prediction of the LQE”. Here, the word “prediction” is wrongly used. “Evaluation” or “estimation” are, in my view, more appropriate since no prediction is done; Please check the entire manuscript for the use of “predict”;

4.        Lines 190-192: the authors wrongly use the term “hierarchical WSN”. They actually speak about a WSN with star network topology. To be more specific, “hierarchical WSN” networks, besides star network topology, may have a cluster-tree topology which is not covered by sentence from lines 190-192;

5.       The authors may generalize their method for ANDON towers to other similar devices.

Author Response

Dear Reviewer

Thank you for giving us the opportunity to submit a revised version of the manuscript “Link Quality Estimation for wireless ANDON towers based on Deep Learning Models” for publication in the Sensor Journal. We appreciate the time and effort that the reviewers dedicated to providing feedback on our manuscript and we are grateful for the insightful comments that led to valuable improvements to our paper.

We have fully revised our manuscript and incorporated the suggestions kindly made by the reviewers. Below we provide, in blue, the point-by-point responses. All page numbers refer to the revised manuscript file with tracked changes enabled.

We hope that the revisions in the manuscript and our accompanying responses will be sufficient to make our manuscript suitable for publication in Sensor Journal.

Reviewer Point:
1. English needs polishing. There are some typos and grammar mistakes. See for examples: a) line 88: it is written “work” instead of “works”; b) line 161: it is written “propose predict LQE”; c) line 249: it is written “In the experimental scenario, Figure 1” instead of ““In the experimental scenario presented in Figure 1”; d) line 314: it is written “epoch” instead of “epochs”; etc. Please check the entire manuscript.

Reply: Thank you for this comment, we have fully revised the entire manuscript, including, but not limited to, typographical and grammar mistakes in lines 88, 171, 260, 326 and 330.

Reviewer Point:
2. Related Work section needs to be extended with a paragraph related to the use of other types of sensor nodes to estimate the link quality within WSN;

Reply: We have added the suggested paragraph in page 3, lines 115 to 124. We describe others types of machine monitoring systems and their relevance to LQE estimation with Machine Learning. We also added reference 23.

Reviewer Point:
3. Line 183: the authors speak about “prediction of the LQE”. Here, the word “prediction” is wrongly used. “Evaluation” or “estimation” are, in my view, more appropriate since no prediction is done; please check the entire manuscript for the use of “predict”;

Reply: Thank you for pointing this out. As suggested by the reviewer, we have changed this in the revised manuscript. Particularly in  lines 130, 310, 317, 440 and 450 the word prediction was replaced by estimation.

Reviewer Point:
4. Lines 190-192: the authors wrongly use the term “hierarchical WSN”. They actually speak about a WSN with star network topology. To be more specific, “hierarchical WSN” networks, besides star network topology, may have a cluster-tree topology which is not covered by sentence from lines 190-192;

Reply: We think this is an excellent point, we truly appreciate it. We have changed this in the revised manuscript, line 200 “In WSN with star network topology,….”

Reviewer Point:
5. The authors may generalize their method for ANDON towers to other similar devices;

Reply: We appreciate the reviewer’s feedback. Although the method we proposed could be generalized to other sensor-based machinery monitoring systems, we consider that it is important to limit the application of our current work to ANDON towers since they are relatively inexpensive devices, frequently used in the industry, and easily adapted to any type of machinery. We also include a new paragraph in the related work section with a description of other types of machine monitoring systems, where our method could be implemented.

Reviewer 3 Report

The main content of research presented in the paper is a deep learning model, suitable for implementation in small workshops with limited computational resources.

The topic is not unique, but it is worthy of researching.

The main proposal is a novel application of a deep learning multilayer perceptron model for the prediction of the predict link quality metric for a wireless sensor network integrated by ANDON towers that send wireless information about alarm signals or the operational status of a workshop's machinery.

The deduced conclusions based on the research methods are that the multilayer perceptron model achieved an accuracy of 99.3% for test data, and shows better performance than other machine learning models such as Support Vector Machines, Random Forest, K-Nearest Neighbors and Gradient boosting.

The conclusions are tenable. The progress that has been made compared to the current survey results has to do with the alarm signal used in conjunction with predict link quality as a novel application for wireless ANDON towers, with the potential to ameliorate problems faced by factories located in  developing countries through the incorporation of new technology during transition to the Industry 4.0 paradigm.

The abstract is informative. It reflects the body of the paper.

The introduction provides sufficient background information for readers in the immediate field to understand the problem.

The text is well arranged and the logic is clear. There are virtually no grammatical errors in this article. However, proofreading the text by a native English speaker would be beneficial to eliminate some stylistic aspects of the text. The related concepts are introduced clearly. The readability is sufficient.

The approaches and techniques used in the study are not new. The novelty lies in its application to a specific situation. A novel application of a deep learning multilayer perceptron model for the prediction of the link quality metric for a wireless sensor network integrated by ANDON towers that send wireless information about alarm signals or the operational status of a workshop's machinery.

The derivation of formulas is rigorous enough.

The theoretical analysis is sufficient for the purposes of the article.

All figures and tables are clear enough to summarize the results for presentation to the readers. All figures and tables are well referred to in the text.

The reference section is informative. However, the formatting of the references in the References section must be revised, completed and corrected to make it more homogeneous and in accordance with the journal's rules.

Author Response

Dear Reviewer

Thank you for giving us the opportunity to submit a revised version of the manuscript “Link Quality Estimation for wireless ANDON towers based on Deep Learning Models” for publication in the Sensor Journal. We appreciate the time and effort that the reviewers dedicated to providing feedback on our manuscript and we are grateful for the insightful comments that led to valuable improvements to our paper.

We have fully revised our manuscript and incorporated the suggestions kindly made by the reviewers. Below we provide, in blue, the point-by-point responses. All page numbers refer to the revised manuscript file with tracked changes enabled.

We hope that the revisions in the manuscript and our accompanying responses will be sufficient to make our manuscript suitable for publication in Sensor Journal.

Reviewer Point:
The main content of research presented in the paper is a deep learning model, suitable for implementation in small workshops with limited computational resources.

The topic is not unique, but it is worthy of researching.

The main proposal is a novel application of a deep learning multilayer perceptron model for the prediction of the predict link quality metric for a wireless sensor network integrated by ANDON towers that send wireless information about alarm signals or the operational status of a workshop's machinery.

The deduced conclusions based on the research methods are that the multilayer perceptron model achieved an accuracy of 99.3% for test data, and shows better performance than other machine learning models such as Support Vector Machines, Random Forest, K-Nearest Neighbors and Gradient boosting.

The conclusions are tenable. The progress that has been made compared to the current survey results has to do with the alarm signal used in conjunction with predict link quality as a novel application for wireless ANDON towers, with the potential to ameliorate problems faced by factories located in  developing countries through the incorporation of new technology during transition to the Industry 4.0 paradigm.

The abstract is informative. It reflects the body of the paper.

The introduction provides sufficient background information for readers in the immediate field to understand the problem.

The text is well arranged and the logic is clear. There are virtually no grammatical errors in this article. However, proofreading the text by a native English speaker would be beneficial to eliminate some stylistic aspects of the text. The related concepts are introduced clearly. The readability is sufficient.

The approaches and techniques used in the study are not new. The novelty lies in its application to a specific situation. A novel application of a deep learning multilayer perceptron model for the prediction of the link quality metric for a wireless sensor network integrated by ANDON towers that send wireless information about alarm signals or the operational status of a workshop's machinery.

The derivation of formulas is rigorous enough.

The theoretical analysis is sufficient for the purposes of the article.

All figures and tables are clear enough to summarize the results for presentation to the readers. All figures and tables are well referred to in the text.

Reply: Thank you for your comments.

Reviewer Point: The reference section is informative. However, the formatting of the references in the References section must be revised, completed and corrected to make it more homogeneous and in accordance with the journal's rules

Reply: Thank you for pointing this out. We have revised the references section and added references 10, 17, whilst verifying that all of them are in accordance with the journal´s rules.

Round 2

Reviewer 2 Report

The authors have successfully solved all my comments/concerns.